# Multi-Perspective Data Augmentation for Few-shot Object Detection

**Anh-Khoa Nguyen Vu**[1,2]  **Quoc-Truong Truong**[1,2]  **Vinh-Tiep Nguyen** [1,2,*]
**Thanh Duc Ngo**[1,2]  **Thanh-Toan Do**[3]  **Tam V. Nguyen**[4]
[1]University of Information Technology, Ho Chi Minh City, Vietnam
[2]Vietnam National University, Ho Chi Minh City, Vietnam
[3]Department of Data Science and AI, Monash University, Australia
[4]University of Dayton, Dayton, OH 45469, United States

## Abstract

Recent few-shot object detection (FSOD) methods have focused on augmenting synthetic samples for novel classes, show promising results to the rise of diffusion models. However, the diversity of such datasets is often limited in representativeness because they lack awareness of typical and hard samples, especially in the context of foreground and background relationships. To tackle this issue, we propose a Multi-Perspective Data Augmentation (MPAD) framework. In terms of foreground-foreground relationships, we propose in-context learning for object synthesis (ICOS) with bounding box adjustments to enhance the detail and spatial information of synthetic samples. Inspired by the large margin principle, support samples play a vital role in defining class boundaries. Therefore, we design a Harmonic Prompt Aggregation Scheduler (HPAS) to mix prompt embeddings at each time step of the generation process in diffusion models, producing hard novel samples. For foreground-background relationships, we introduce a Background Proposal method (BAP) to sample typical and hard backgrounds. Extensive experiments on multiple FSOD benchmarks demonstrate the effectiveness of our approach. Our framework significantly outperforms traditional methods, achieving an average increase of $17.5\%$ in nAP50 over the baseline on PASCAL VOC. Code is available at github.com/nvakhoa/MPAD.

## 1 Introduction

Humans can recognize new objects after seeing them just a few times, a remarkable ability that is simulated and studied in few-shot object detection (FSOD). In an FSOD setup, there are two distinct datasets: the base dataset and the novel dataset. The base dataset is extensive and comprises numerous classes with abundant training instances. This dataset helps the model learn a wide variety of object features and characteristics, forming a general knowledge for detection tasks. In contrast, the novel dataset is limited, with only a few samples per novel class, posing a significant challenge for object detection. This constraint makes FSOD a critical research area (Yan et al., 2019; Kang et al., 2019; Fan et al., 2020; Wang et al., 2020; Li et al., 2021; Li & Li, 2021b; Zhang et al., 2021; Han et al., 2022b; Bulat et al., 2023), with potential applications in fields such as robotics, autonomous driving, and medical imaging, where models need to handle critical but rare scenarios.

Earlier FSOD approaches (Wang et al., 2020; Qiao et al., 2021; Yan et al., 2019; Zhang et al., 2021) firstly train a model on the base dataset to establish a generalized detector. This detector is then fine-tuned on the novel dataset to recognize and detect new objects. Still, this approach could lead to overfitting due to the limited amount of data available. Other methods (Zhu et al., 2021; Li et al., 2023a) leverage the general knowledge of the large language models (LLMs) to alleviate this issue. A simply yet effective approach for FSOD is data augmentation. Recent works (Zhang & Wang, 2021; Vu et al., 2023b) utilize the prior knowledge to create hallucinations in feature space to fine-tune classifiers. However, these synthetic samples often lack essential information for object detection, such as low level details, spatial information. Meanwhile, other methods (Li & Li, 2021a;

---

*Corresponding Authors: tiepnv@uit.edu.vn

Demirel et al., 2023) rely solely on traditional geometric transformations (e.g., flipping, cropping, rotating) to create variations of given samples from novel classes, which limits the diversity of synthesized datasets.

Recently, diffusion models have achieved remarkable strides in producing high-quality and diverse datasets (Nichol et al., 2021; Rombach et al., 2022; Ramesh et al., 2022; Saharia et al., 2022). Furthermore, large-scale text-to-image diffusion models have shown significant flexibility and scalability in image editing tasks by incorporating lightweight adapter modules for additional conditions (e.g., bounding box, semantic map, depth map, human pose) (Zhang et al., 2023; Li et al., 2023b; Zhuang et al., 2023). Consequently, several FSOD methods (Lin et al., 2023; Fang et al., 2024; Wang et al., 2024) leverage controllable diffusion but often reply on simple prompts to generate synthetic objects, without exploring attributes such as colors, shapes, details, sizes, types of objects. As a result, most synthesized novel samples are typical objects.

To address the above problem, we propose In-Context learning for Object Synthesis (ICOS). ICOS leverages general knowledge from LLMs to deeply explore the attributes of novel classes and diversify prompt inputs. Additionally, the diversity of a class is derived from both ***typical*** and ***hard*** samples, as illustrated in Figure 1. Inspired by the large margin principle (e.g. SVM (Cortes, 1995)), support vectors play a crucial role in learning a generalized model. These samples, considered hard samples, often exhibit characteristics not only of the main class but also of neighboring ones. In other words, in this paper, we define typical samples as those that contain features of a single class, whereas hard samples exhibit features of two classes. Leveraging this aspect, we aim to blend the characteristics of two classes during the data generation process. Unlike image classification, where only the foreground-foreground relations are considered and the main objects are roughly centered, object detection must take into account the foreground-background relations. To our knowledge, this is the first work to use ChatGPT to diversify prompts and embed the foreground-background relations when synthesizing diverse datasets in few-shot object detection.

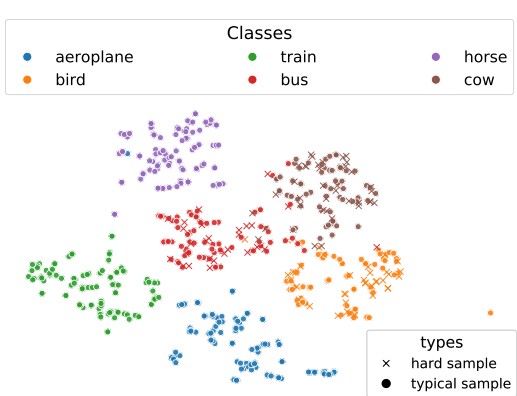

Figure 1: T-SNE visualization of novel synthetic samples and base real samples in Novel Set 1 of PASCAL VOC. We only generate synthetic samples for three novel classes ("bird", "bus", "cow") and use real samples for three base classes ("aeroplane", "train", "horse"). Typical and hard samples in novel classes are created by using ICOS and HPAS, respectively. Base real samples are considered as typical samples.

In terms of the *foreground-foreground* relation, we propose a Harmonic Prompt Aggregation Scheduler (HPAS) to mix prompt embeddings at each time step of the generation process in the diffusion model. This approach guides the diffusion model to synthesize objects with high-level features (e.g., object parts) of the main class and low-level features (e.g., shape, color, size) of a selected base class. By mixing the low-level features of the base class, we leverage the prior knowledge acquired during the base training stage. Regarding *foreground-background* relation, we introduce a Background Proposal method (BAP) to sample typical and hard backgrounds from the base dataset. For typical backgrounds, we select the most cluttered backgrounds based on an entropy metric. For hard backgrounds, we select those with the highest similarity to foreground objects in the embedding space. During the base training stage, the model learns to classify novel objects as backgrounds when trained on base classes. This phenomenon creates ambiguities in learning and detecting novel classes. Therefore, in the novel training stage, we guide the model to distinguish novel classes from similar base backgrounds, utilizing the knowledge gained from base training.

In summary, our contributions can be summarized as follows:

- We propose a Multi-Perspective Data Augmentation (MPAD) framework for synthesizing data which better prevents the overfitting problem for FSOD.

- We introduce ICOS, HPAS, and BAP methods to enhance synthesis by considering foreground-background relation. Specifically, ICOS diversifies prompts using fine-grained attributes from the general knowledge of LLMs. HPAS supports the controllable diffusion model to create hard samples containing characteristics of two foregrounds, while BAP proposes typical and hard backgrounds in relation to the foreground.

- We conduct comprehensive experiments on FSOD benchmarks to demonstrate the effectiveness of our method. The results show that our method outperforms the baseline model by a large margin and achieves state-of-the-art performance on few-shot object detection.

## 2 MPAD METHOD

### 2.1 FORMULATION

In few-shot object detection, the base data is characterized by a large number of base classes $C_{base}$ with an abundance of samples. In contrast, the novel data comprises a few novel classes $C_{novel}$, each with $K$ samples ($K \in \{1, 2, 3, 5, 10\}$ in the PASCAL VOC setting). It is important to note that the base classes and novel classes are disjoint sets (i.e., $C_{base} \cap C_{novel} = \emptyset$). As outlined in previous works (Yan et al., 2019; Wang et al., 2020; Qiao et al., 2021), we define two data sets $D_s = \{(I_s^i, A_s^i)\}_{i=1..N_s}$, where $s \in \{base, novel\}$. $I_s$, $A_s$ and $N_s$ denote the images, annotations and number of samples in set $s$, respectively. An annotation $A_s^{i,j} = (c, b)$ represents a pair consisting of a class name $c \in C_s$ and the bounding box $b$ of the $j$-th object in the $i$-th image.

Typically, FSOD methods involve two stages: base training stage and novel fine-tuning stage. In the base training stage, detectors are trained on $D_{base}$ to acquire extensive knowledge, learn concept features, and build the feature extractor . In the novel fine-tuning stage, the base models are fine-tuned on a balanced set $D_{ft}$ with $K$ samples for each base and novel class to detect both base and novel objects in the image.

### 2.2 FOREGROUND-BACKGROUND RELATION-AWARE DATA AUGMENTATION

In object detection, an image comprises two main components: the background and the foreground. The foreground highlights the primary objects, while the background provides contextual information that aids in object inference within the images. To augment data with class representativeness, we synthesize both typical and hard samples. For typical foreground samples, ICOS uses input samples to generate novel objects with characteristics pointed out by general knowledge of LLMs. To create hard samples, HPAS mixes prompt embeddings at each time step of the data generation process in the diffusion model. Different from the classification task, the background plays an important role in object detection. Therefore, BAP proposes hard background samples in relation to foreground features. As shown in Figure 2, our overall framework contains three main components: ICOS, HPAS, and BAP, as detailed in the following subsections.

### 2.3 IN-CONTEXT LEARNING FOR OBJECT SYNTHESIS

**Controllable diffusion.** We utilize PowerPaint (Zhuang et al., 2023) model for the object inpainting task, ensuring that the generated object seamlessly conforms to the specified mask shape. We process the object's bounding box by applying masking and padding, and then use it as the mask input for controllable diffusion. The controllable diffusion $\theta(\cdot)$ takes a prompt embedding $\zeta_c$, a bounding box $b$, and an image $I$ as inputs. The reverse diffusion process is a sequence of denoising steps with time step $t = T, T-1, \ldots, 1$.

$$z_{t-1} = p(z_{t-1}|\theta(z_t, \zeta_c, b)), \tag{1}$$

where $z_T$ is the reference image $I$ and $z_0$ is the synthesized image $\hat{I}$. For each novel class $c \in C_{novel}$, we generate $N$ synthesis samples. We add novel objects to random base images $I_{base}$, defined as:

$$\hat{I}_c = \theta(I_{base}^i, \zeta_c, b), \tag{2}$$

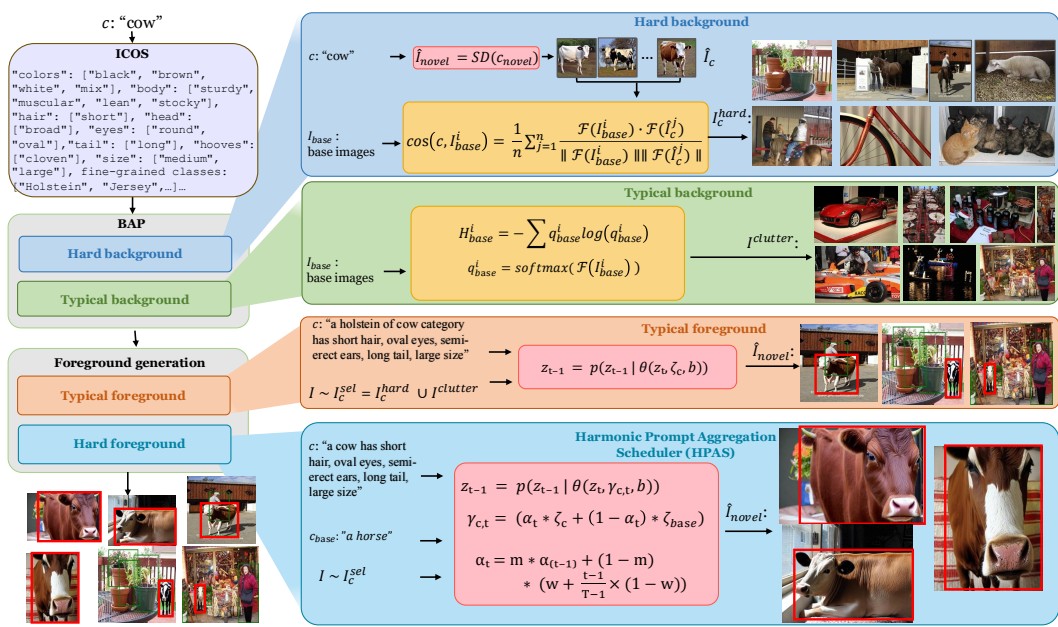

Figure 2: The overall framework. To exploit the ability of controllable diffusion model for FSOD, we proposed a novel data augmentation method that incorporates various aspects to generate diverse data. Our method includes ICOS, BAP, HPAS. ICOS aims to deeply explore the attributes of novel classes and diversify the prompt for controllable diffusion models. BAP selects hard and typical backgrounds while HPAS generates hard (mixed) instances

.

where $i \sim U(1, N_{base})$, $\zeta_c = \mathcal{E}(\texttt{prompt}_c)$ is the prompt embedding and $\mathcal{E}$ is the CLIP text encoder. The bounding box $b$ is randomly chosen from the annotations of $\hat{I}^i_{base}$ unless otherwise specified. The class label of the selected bounding box $b$ is replaced by $c$.

**Simple Prompting.** Following previous works (Lin et al., 2023; Fang et al., 2024; Wang et al., 2024), a simple prompt is created by concatenating the prefix "a" and the class name, resulting in the simple prompt input $\texttt{prompt}_c =$ "a photo of a [CLASSNAME]".

**In-Context learning for Object Synthesis (ICOS).** The simple prompt mentioned above only outlines a general concept of the object without detailed information, which can result in similar objects within a class and limit diversity. To address this, we propose using in-context learning (Reynolds & McDonell, 2021) to collect and incorporate specific characteristics and class information, enhancing the diversity of the prompts for the diffusion model.

*In-Context learning for Attribute Analysis.* Based on a recent work (Zhu et al., 2024), we explore the attributes of a specific class using LLMs. Specifically, we construct an input and output template to extract appearance information of a class using ChatGPT. Figure 3 (a) demonstrates a in-context learning approach for analyzing parts and attribute values of a class, where the target class name is input for the next inference. We then parse the attributes into a dictionary, with keys and values representing the general appearances and detailed attributes of the class. We randomly select a key-value attributes list $[\texttt{attr}] = \{\texttt{key}_i, \texttt{value}_i\}_{i=1..n_a}$ to additional provide information and diversify the prompt. Specifically, we construct the new prompt from $[\texttt{attr}]$ by the template as $\texttt{prompt}_c =$ "a [CLASS NAME] has [$\texttt{key}_1$] [$\texttt{value}_1$], [$\texttt{key}_2$] [$\texttt{value}_2$],..., [$\texttt{key}_{n_a}$] [$\texttt{value}_{n_a}$]".

*In-Context learning for Fine-Grained Categories.* Fine-grained categories are crucial for assessing the diversity within a class. Several methods (Vu et al., 2023a; Wu et al., 2024) exploit this aspect to improve model generalization. SMS (Vu et al., 2023a) introduces a technique that utilizes fine-grained categories in few-shot instance segmentation by generating hallucinated superclasses from base and novel classes. Inspired by SMS, we leverage LLMs by querying ChatGPT to list the fine-grained categories of class $c$ using the prompt illustrated in Figure 3 (b).

**In-Context learning for attribute analysis**

Describe the features and attributes of each class after we enter the input class. With each class, return the response, and attribute following this instruction:

**Input sample:**
Describe the appearance attributes of cow categories.

**Ouput sample:**
`response:` Cows typically have colors such as black, white, brown, or a combination of these. Their fur is usually short and coarse...
`atrr: {`
`"cow" : {`
`    "colors": ["black", "white", "brown"],`
`    "fur": ["short", "coarse"],`
`    "body": ["large", "rounded", "stocky"],`
`    "ears": ["medium-sized", "floppy"],..`
`},..}`

**Input:** [CLASS NAME]

**In-Context learning for fine-grained objects**

List the fine-grained categories within an object class. With each class, return attribute following this instruction:

**Input sample:** cow

**Output sample:**
`atrr: {`
`"cow": {`
`    "Fine-grained classes":["Holstein Friesian", "Jersey", "Guernsey", "Brown Swiss", "Ayrshire", "Hereford", "Angus", "Charolais", "Limousin", "Simmental", "Brahman", "Gelbvieh", "Shorthorn", "Texas Longhorn", "Highland", "Red Poll", "Devon", "Galloway", "Belgian Blue", "Watusi"]`
`},..}`

**Input:** [CLASS NAME]

(a)                                                    (b)

Figure 3: In-context learning technique for exploring (a) attributes and (b) fine-grained object categories of a novel class given a sample. The input [CLASSNAME] is replaced by class name $c \in C_{novel}$.

The result of this query is parsed and added to [attr] to generate a diverse set of prompts. The final $\text{prompt}_c$ is randomly sampled from the attribute list [attr] and then used for synthesizing novel class samples. See Figure 6 and Figure 7 in Appendix A for detailed responses of ICOS.

## 2.4 HARMONIC PROMPT AGGREGATION SCHEDULER

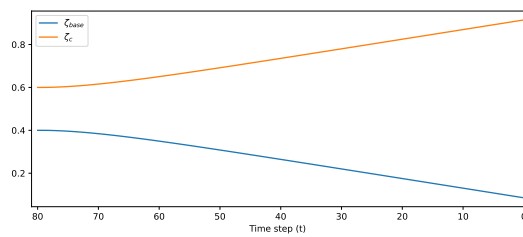

Figure 4: Visualization of the weighted values of the Harmonic Prompt Aggregation Scheduler across the timesteps of controllable diffusion.

In addition to leveraging hard novel samples for the few-shot object detection model, we introduce a mechanism called Harmonic Prompt Aggregation Scheduler (**HPAS**). The main idea is to mix a base class with a similar novel class to enhance the diversity of the synthetic dataset. This is achieved in the prompt embedding space, step by step, throughout the generation process of the diffusion model. The prompt embedding aggregation scheduler is defined:

$$\gamma_{c,t} = (1 - \alpha_t) * \zeta_c + \alpha_t * \zeta_{base}, \quad (3)$$

where $\alpha_t = m \times \alpha_{(t-1)} + (1 - m) \times \left((w + \frac{t-1}{T-1} \times (1-w)\right)$, $t = 1, \ldots, T$. $\alpha_t$ is the weighted value at $t$-th time step and $w$ is the starting value. By gradually increasing the weight of novel class features and reducing that of the base class, we create a synthetic object that incorporates novel detailed characteristics within the low-level features of the base class. Inspired by He et al. (2020), the momentum $m$ is used to retain the main features of the prompt embedding. The weighted values are shown in Figure 4. We substitute $\gamma_{c,t}$ from Eq. (3) into Eq. (1). In this way, we can create hard samples, as shown in Figure 5. The reverse diffusion process of the diffusion model becomes:

$$z_{t-1} = p(z_{t-1}|\theta(z_t, \gamma_{c,t}, b)) \quad (4)$$

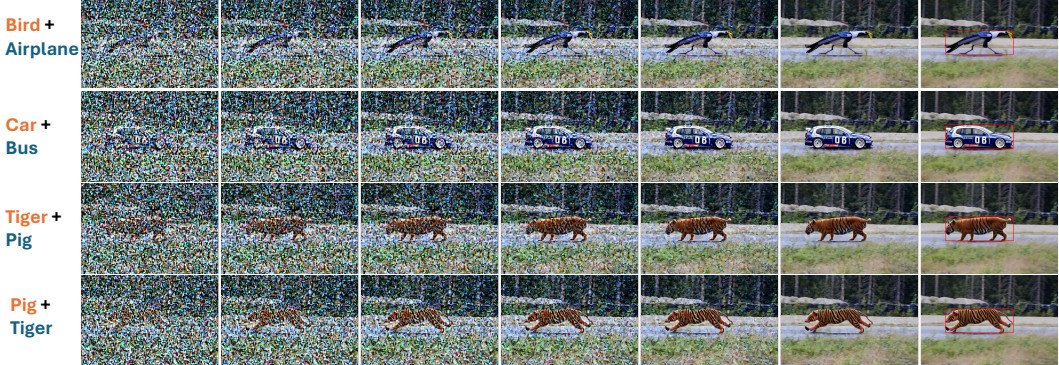

Figure 5: Visualization of the mixed instances of the Harmonic Prompt Aggregation Scheduler during the generation data process in the controllable diffusion model.

## 2.5 BACKGROUND PROPOPOSAL

The background plays an important role in object detection tasks, where the model must distinguish not only between foreground objects, but also between foreground and background. In the base training stage, the model is trained to classify novel classes as background due to the condition $C_{base} \cap C_{novel} = \emptyset$. Therefore, in the novel training stage, we need to guide the model to efficiently distinguish novel classes from new backgrounds by utilizing backgrounds with similar visual features. To address this issue, we introduce the background proposal (**BAP**), which includes both the hard background proposals and the typical background proposals.

**Hard background proposal.** Inspired by Le et al. (2019) where objects concealing in the backgrounds, these camouflaged objects have a foreground that visually resembles the background and it creates difficulties for the model when detecting them. Therefore, we introduce a visual similarity background technique to create hard samples.

We select backgrounds from base images $I_{base}$ that share similar features with the novel class $c$ by employing cosine similarity. Instead of using textual embeddings, which may not capture essential visual information, we use a pretrained visual encoder $\mathcal{F}(\cdot)$ (e.g., ViT (Dosovitskiy et al., 2021)). In FSOD, the number of novel samples is insufficient to represent the general class distribution. Therefore, we use the stable diffusion model (Rombach et al., 2022) $SD(\cdot)$ to synthesize a set of samples for class $c$, denoted by $\hat{I}_c$. This synthetic set is used for selecting hard backgrounds. The cosine similarity metric is defined as follows:

$$\cos(c, I_{base}^i) = \frac{1}{n} \sum_{j=0}^{n} \frac{\mathcal{F}(I_{base}^i) \cdot \mathcal{F}(\hat{I}_c^j)}{\|\mathcal{F}(I_{base}^i)\| \|\mathcal{F}(\hat{I}_c^j)\|} \tag{5}$$

where $\hat{I}_c = \left\{ \hat{I}_c^j \mid \hat{I}_c^j = SD(\text{prompt}_c) \right\}_{j=1}^{n}$ are synthesized images of class $c$. We select the top base backgrounds with the highest similarity scores to the novel class $c$, denoted by $I_c^{hard}$.

**Typical clutter background.** For the typical background, we sample from $I_{base}$ ones that have clutter features representing scenes with crowded and complex environments (as defined in Rosenholtz et al. (2007)). These samples with noise features force the model to improve its localization ability.

In this paper, we use the entropy score to quantify the clutter level of an image. Specifically, we normalize the feature embedding of the image using the softmax function. Then, we apply the entropy formula as follows:

$$H_{base}^i = -\sum q_{base}^i \log(q_{base}^i), \tag{6}$$

| Method | Novel Set 1 | | | | | Novel Set 2 | | | | | Novel Set 3 | | | | | Mean |
|---|---|---|---|---|---|---|---|---|---|---|---|---|---|---|---|---|
| | 1 | 2 | 3 | 5 | 10 | 1 | 2 | 3 | 5 | 10 | 1 | 2 | 3 | 5 | 10 | |
| TFA w/ fc (Wang et al., 2020) | 22.9 | 34.5 | 40.4 | 46.7 | 52.0 | 16.9 | 26.4 | 30.5 | 34.6 | 39.7 | 15.7 | 27.2 | 34.7 | 40.8 | 44.6 | 33.8 |
| TFA w/ cos (Wang et al., 2020) | 25.3 | 36.4 | 42.1 | 47.9 | 52.8 | 18.3 | 27.5 | 30.9 | 34.1 | 39.5 | 17.9 | 27.2 | 34.3 | 40.8 | 45.6 | 34.7 |
| FSDetView (Xiao & Marlet, 2020) | 24.2 | 35.3 | 42.2 | 49.1 | 57.4 | 21.6 | 24.6 | 31.9 | 37.0 | 45.7 | 21.2 | 30.0 | 37.2 | 43.8 | 49.6 | 36.7 |
| MPSR (Wu et al., 2020) | 41.7 | 42.5 | 51.4 | 55.2 | 61.8 | 24.4 | 29.3 | 39.2 | 39.9 | 47.8 | 35.6 | 41.8 | 42.3 | 48.0 | 49.7 | 43.4 |
| FSCE (Sun et al., 2021) | 32.9 | 44.0 | 46.8 | 52.9 | 59.7 | 23.7 | 30.6 | 38.4 | 43.0 | 48.5 | 22.6 | 33.4 | 39.5 | 47.3 | 54.0 | 41.2 |
| SRR-FSD (Zhu et al., 2021) | 47.8 | 50.5 | 51.3 | 55.2 | 56.8 | 32.5 | 35.3 | 39.1 | 40.8 | 43.8 | 40.1 | 41.5 | 44.3 | 46.9 | 46.4 | 44.8 |
| DCNet (Hu et al., 2021) | 33.9 | 37.4 | 43.7 | 51.1 | 59.6 | 23.2 | 24.8 | 30.6 | 36.7 | 46.6 | 32.3 | 34.9 | 39.7 | 42.6 | 50.7 | 39.2 |
| Meta DETR (Zhang et al., 2021) | 49.0 | 53.2 | 57.4 | 62.0 | 27.9 | 32.3 | 38.4 | 43.2 | 51.8 | 34.9 | 41.8 | 47.1 | 54.1 | 58.2 | 45.8 | 45.8 |
| Meta F R-CNN (Han et al., 2022a) | 43.0 | 54.5 | 60.6 | 66.1 | 65.4 | 27.7 | 35.5 | 46.1 | 47.8 | 51.4 | 40.6 | 46.4 | 53.4 | 59.9 | 58.6 | 50.5 |
| KFSOD (Zhang et al., 2022) | 44.6 | - | 54.4 | 60.9 | 65.8 | 37.8 | - | 43.1 | 48.1 | 50.4 | 34.8 | - | 44.1 | 52.7 | 53.9 | 49.2 |
| DeFRCN (Qiao et al., 2021) | 40.2 | 53.6 | 58.2 | 63.6 | 66.5 | 29.5 | 39.7 | 43.4 | 48.1 | 52.8 | 35.0 | 38.3 | 52.9 | 57.7 | 60.8 | 49.4 |
| MFD (Wu et al., 2022) | 63.4 | 66.3 | 67.7 | 69.4 | 68.1 | 42.1 | 46.5 | 53.4 | 55.3 | 53.8 | 56.1 | 58.3 | 59.0 | 62.2 | 63.7 | 59.0 |
| FCT (Han et al., 2022b) | 57.1 | 57.9 | 63.2 | 67.1 | 27.6 | 34.5 | 43.7 | 49.2 | 51.2 | 39.5 | 54.7 | 52.3 | 57.0 | 58.7 | 50.9 | 50.0 |
| FS-DETR (Bulat et al., 2023) | 45.0 | 48.5 | 51.5 | 52.7 | 56.1 | 37.3 | 41.3 | 43.4 | 46.6 | 49.0 | 43.8 | 47.1 | 50.6 | 52.1 | 56.9 | 48.1 |
| D&R (Li et al., 2023a) | 41.0 | 51.7 | 55.7 | 61.8 | 65.4 | 30.7 | 39.0 | 42.5 | 46.6 | 51.7 | 37.9 | 47.1 | 51.7 | 56.8 | 59.5 | 49.3 |
| VFA (Han et al., 2023) | 47.4 | 54.4 | 58.5 | 64.5 | 66.5 | 33.7 | 38.2 | 43.5 | 48.3 | 52.4 | 43.8 | 48.9 | 53.3 | 58.1 | 60.0 | 51.4 |
| FSRN (Guirguis et al., 2023a) | 19.7 | 33.9 | 42.3 | 51.9 | 55.1 | 18.5 | 24.7 | 27.3 | 35.2 | 47.5 | 26.7 | 37.0 | 41.2 | 47.5 | 51.7 | 37.3 |
| DiGeo (Ma et al., 2023) | 37.9 | 39.4 | 48.5 | 58.6 | 61.5 | 26.6 | 28.9 | 41.9 | 42.1 | 49.1 | 30.4 | 40.1 | 46.9 | 52.7 | 54.7 | 44.0 |
| NIFF (Guirguis et al., 2023b) | 46.0 | 57.2 | 62.0 | 65.5 | 67.2 | 30.1 | 39.6 | 45.0 | 49.4 | 52.8 | 41.1 | 52.5 | 56.4 | 59.7 | 62.1 | 52.4 |
| Using deep learning augmentation technique | | | | | | | | | | | | | | | | |
| TIP (Li & Li, 2021a) | 27.7 | 36.5 | 43.3 | 50.2 | 59.6 | 22.7 | 30.1 | 33.8 | 40.9 | 46.9 | 21.7 | 30.6 | 38.1 | 44.5 | 50.9 | 38.5 |
| Halluc (Zhang & Wang, 2021) | 47.0 | 44.9 | 46.5 | 54.7 | 54.7 | 26.3 | 31.8 | 37.4 | 37.4 | 41.2 | 40.4 | 42.1 | 43.3 | 51.4 | 49.6 | 43.2 |
| LVC (Kaul et al., 2022) | 54.5 | 53.2 | 58.8 | 63.2 | 65.7 | 32.8 | 29.2 | 50.7 | 49.8 | 50.6 | 48.4 | 52.7 | 55 | 59.6 | 59.6 | 52.3 |
| Norm-VAE (Xu et al., 2023) | 62.1 | 64.9 | 67.8 | 69.2 | 67.5 | 39.9 | 46.8 | 54.4 | 54.2 | 53.6 | 58.2 | 60.3 | 61.0 | 64.0 | 65.5 | 59.3 |
| SFOT (Vu et al., 2023b) | 47.9 | 60.4 | 62.7 | 67.3 | 69.1 | 32.4 | 41.2 | 45.7 | 50.2 | 54.0 | 43.5 | 54.1 | 56.9 | 60.6 | 62.5 | 53.9 |
| Lin et al. (Lin et al., 2023)† | 67.5 | - | 69.8 | 71.1 | 71.5 | 52.0 | - | 54.3 | 57.5 | 57.4 | 55.9 | - | 58.6 | 59.6 | 63.9 | 61.6 |
| SNIDA (Wang et al., 2024) | 59.3 | 60.8 | 64.3 | 65.4 | 65.6 | 35.2 | 40.8 | 50.2 | 54.6 | 50.0 | 51.6 | 52.4 | 55.9 | 58.5 | 62.6 | 55.1 |
| **MPAD** | **69.1** | **69.5** | 69.6 | 69.9 | 68.9 | **58.4** | **59.7** | **61.8** | **61.8** | **63.5** | **70.1** | **69.8** | **69.9** | **70.4** | **71.4** | **66.7** |

Table 1: Generalized few-shot object detection performance (nAP50) on the PASCAL VOC dataset. The best and second performances are marked in **boldface** and underlined, respectively. † indicates using post-detection process.

where $q_{base}^i = \mathrm{softmax}(\mathcal{F}(I_{base}^i))$ and $H_{base}^i$ are the feature distribution and information entropy of the background image $I_{base}^i$, respectively. We select the top base backgrounds with the highest entropy score, denoted by $I^{clutter}$.

## 2.6 DATA GENERATION AND MODEL TRAINING PROCESS

In summary, for each novel class $c$, we use the proposed backgrounds $I_c^{sel} = I_c^{hard} \cup I^{clutter}$ to synthesize novel images $\hat{I}_{novel}$. We generate two types of foregrounds: the typical foreground and the hard foreground. For typical foregrounds, we use in-context learning with diverse attributes and fine-grained categories to create $\texttt{prompt}_c$. Then, images with the novel class $c$ are then synthesized through the reverse diffusion process conditioned on prompt embedding $\zeta_c$, as illustrated in Eq. (1). For hard foregrounds, we use Eq. (4) to generate mixed instances between class $c$ and a selected base class. The annotation $\hat{A}_{novel}^i$ for a synthesized image $\hat{I}_{novel}^i$ is copied from the original annotation data of the selected base image, with the class name within the bounding box $b$ replaced by $c$. The novel synthetic dataset $\hat{D}_{novel} = \{(\hat{I}_{novel}^i, \hat{A}_{novel}^i)\}_{i=1\dots\hat{N}_{aug}} \cup D_{ft}$ is used to fine-tune detectors during the novel training stage. See Figure 8 in Appendix B for additional visualizations of our synthetic dataset.

## 3 EXPERIMENTS

### 3.1 DATASETS AND SETTINGS

**Dataset settings and evaluation.** Following previous works (Yan et al., 2019; Kang et al., 2019; Qiao et al., 2021), we assess our MPAD method in the FSOD setting of PASCAL VOC (Everingham et al., 2010; 2015) and MS COCO (Lin et al., 2014). For PASCAL VOC, 20 classes are separated into three sets. In each set, five classes are designated as novel classes $C_{novel}$, and the remaining fifteen classes are used as the base classes $C_{base}$. There are $K$ samples for each novel class ($K \in \{1, 2, 3, 5, 10\}$). Regarding MS COCO, the dataset serves as a challenging benchmark for FSOD. 80 classes are split into 60 base classes and 20 novel classes (identical to the 20 PASCAL VOC classes). We select a value of $K$ from the set ($\{1, 2, 3, 5\}$) for each novel and base class to fine-tune detectors. To evaluate the model performance, we follow TFA (Wang et al., 2020), DeFRCN (Qiao et al., 2021) and use the Generalized Few-Shot Object Detection (G-FSOD) which contains both base and novel classes to train and test models in the novel fine-tuning stage. We report AP50 of

| Method | 1-shot | | | 2-shot | | | 3-shot | | | 5-shot | | |
|---|---|---|---|---|---|---|---|---|---|---|---|---|
| | nAP | nAP50 | nAP75 | nAP | nAP50 | nAP75 | nAP | nAP50 | nAP75 | nAP | nAP50 | nAP75 |
| TFA w/ fc (Wang et al., 2020) | 1.6 | 3.4 | 1.3 | 3.8 | 7.8 | 3.2 | 5.0 | 9.9 | 4.6 | 6.9 | 13.4 | 6.3 |
| TFA w/ cos (Wang et al., 2020) | 1.9 | 3.8 | 1.7 | 3.9 | 7.8 | 3.6 | 5.1 | 9.9 | 4.8 | 7.0 | 13.3 | 6.5 |
| MPSR (Wu et al., 2020) | 2.3 | 4.1 | 2.3 | 3.5 | 6.3 | 3.4 | 5.2 | - | - | 6.7 | - | - |
| FSDetView (Xiao & Marlet, 2020) | 3.2 | 8.9 | 1.4 | 4.9 | 13.3 | 2.3 | 6.7 | 18.6 | 2.9 | 8.1 | 20.1 | 4.4 |
| DeFRCN (Qiao et al., 2021) | 4.8 | 9.5 | 4.4 | 8.5 | 16.3 | 7.8 | 10.7 | 20.0 | 10.3 | 13.5 | 24.7 | 13.0 |
| Meta-DETR (Zhang et al., 2021) | 7.5 | 12.5 | 7.7 | - | - | - | 13.5 | 21.7 | 14 | 15.4 | 25 | 15.8 |
| FCT (Han et al., 2022b) | 5.6 | - | - | 7.9 | - | - | 11.1 | - | - | 14 | - | - |
| AirDet (Li et al., 2022) | 6.1 | 11.4 | 6.0 | 8.7 | 16.2 | 8.4 | 10.0 | 19.4 | 9.1 | 10.8 | 20.8 | 10.3 |
| Meta F R-CNN (Han et al., 2022a) | 5.1 | 10.7 | 4.3 | 7.6 | 16.3 | 6.2 | 9.8 | 20.2 | 8.2 | 10.8 | 22.1 | 9.2 |
| D&R (Li et al., 2023a) | 6.1 | - | - | 9.5 | - | - | 11.5 | - | - | 13.9 | - | - |
| FSRN (Guirguis et al., 2023a) | - | - | - | - | - | - | - | - | - | 8.7 | 16.1 | 8.2 |
| FS-DETR (Bulat et al., 2023) | 7.0 | 13.6 | 7.5 | 8.9 | 17.5 | 9.0 | 10.0 | 18.8 | 10.0 | 10.9 | 20.7 | 10.8 |
| Using deep learning augmentation technique | | | | | | | | | | | | |
| Halluc (Zhang & Wang, 2021) | 4.4 | 7.5 | 4.9 | 5.6 | 9.9 | 5.9 | 7.2 | 13.3 | 7.4 | - | - | - |
| SFOT (Vu et al., 2023b) | 6.7 | 13.2 | 6.0 | 10.5 | 20.3 | 9.7 | 12.5 | 23.6 | 11.8 | 14.9 | 27.8 | 14.2 |
| Norm-VAE (Xu et al., 2023) | 9.5 | - | 8.8 | 13.7 | - | 13.7 | 14.3 | - | 14.4 | 15.9 | - | 15.3 |
| SNIDA(Wang et al., 2024) | 9.3 | - | - | 12.9 | - | - | 14.8 | - | - | 16.1 | - | - |
| MPAD | 18.3 | 31.2 | 18.8 | 18.5 | 31.6 | 18.9 | 18.8 | 31.8 | 19.1 | 18.9 | 32.4 | 19.3 |

Table 2: Generalized few-shot object detection performance on 1, 2, 3, 5-shot of MS COCO dataset. The best and second performances are marked in **boldface** and underlined, respectively.

novel classes (nAP50) on PASCAL VOC dataset and nAP, nAP50, nAP75 metrics for experiments on the COCO dataset.

**Implementation details.** Our model adopts DeFRCN (Qiao et al., 2021). In both the base training and the fine-tuning stage, we use the same hyper-parameters as DeFRCN (Qiao et al., 2021). During the fine-tuning stage, we utilize both real novel data and synthetic data to train models on a single NVIDIA GeForce RTX 2080 Ti GPU. We employ Powerpaint (Zhuang et al., 2023) for the conditional diffusion model $\theta(\cdot)$, and the CLIP text encoder (Radford et al., 2021) for $\mathcal{E}(\cdot)$. In our experiments, we aim to generate an equal number of synthetic instances for each novel class. The image feature extractor $\mathcal{F}(\cdot)$ is a pre-trained ViT model (Dosovitskiy et al., 2021) on ImageNet (Deng et al., 2009). We set $w = 0.7$, $m = 0.8$ and $\hat{N}_{aug} = 300$. The number of inference steps is fixed at $T = 80$. Several methods show that training with multi-scale objects is crucial in FSOD. Therefore, we implement a fundamental method to increase the diversity under this aspect. In particular, we scale the selected bounding box in the data generation process with a weight. We randomly select weight value in $\{1.25, 1.5, 1.75, 2\}$ in our settings.

## 3.2 RESULTS AND DISCUSSION

### 3.2.1 MAIN RESULTS

We conduct G-FSOD experiments on PASCAL VOC (Everingham et al., 2010; 2015) and MS COCO (Lin et al., 2014) and report the results in Table 1 and Table 2, respectively. These numbers indicate that our method MPAD generally outperforms the baseline and other state-of-the-art methods on FSOD benchmarks by a large margin.

**Results on PASCAL VOC.** Table 1 shows the results from the three novel sets of PASCAL VOC, comparing our approach with baselines and state-of-the-art methods. Our MPAD method consistently outperforms the baselines across all splits and shots. Notably, our method, based on De-FRCN (Qiao et al., 2021), achieves the highest performance of 66.9%, exceeding the baseline by an average margin of 17.5%. In extremely low-shot scenarios, our method delivers significantly larger performance gains, with an average increase of +31.0% nAP50 in the 1-shot setting. Considerably, our MPAD surpasses previous works (Wang et al., 2024; Lin et al., 2023; Kaul et al., 2022; Li et al., 2023a; Zhu et al., 2021; Xu et al., 2023) that use pretrained CLIP, ViT, diffusion models, language models, or post-processing in detection. Meanwhile, methods (Wang et al., 2024; Lin et al., 2023) are state-of-the-art data augmentation methods in FSOD. Overall, our approach demonstrates superior performance compared to most existing methods across various splits and shots, highlighting the robustness and generalization capabilities of our method.

**Results on MS COCO.** We present the experimental results for MS COCO in Table 2. By using our method, baseline DeFRCN improves by about 11.5% on average, particularly in extremely few-shot settings (1 and 2-shot). Specifically, our method enhances nAP, nAP50, and nAP75 by over 13%,

|           | 1-shot | 2-shot | 3-shot | 5-shot | 10-shot | Mean |
|-----------|--------|--------|--------|--------|---------|------|
| Cutout    | 54.7   | 57.2   | 62.3 | 64.0 | 62.5 | 60.1 |
| GridMask  | 54.3   | 58.0 | 62.0 | 63.7 | 63.2 | 60.2 |
| AutoAugment | 51.3 | 54.4   | 59.6   | 62.1   | 61.0    | 57.7 |
| CutMix    | 55.5 | 57.6 | 61.4 | 63.9 | 63.5 | 60.4 |
| MPAD      | **69.1** | **69.5** | **69.6** | **69.9** | **68.9** | **69.4** |

Table 3: Few-shot object detection performance (nAP50) of other augmentation methods on Novel Set 1 of PASCAL VOC dataset. The best and second performances are marked in **boldface** and underlined, respectively.

21%, and 14%, respectively, in the 1-shot setting. These results highlight the promising approach to improving FSOD performance by employing controllable diffusion model.

**Comparison with different augmentation methods.** Following Wang et al. (2024), we also show few-shot object detection results on PASCAL VOC Novel Set 1 of other augmentation methods Cutout (DeVries, 2017), GridMask (Chen et al., 2020), AutoAugment (Zoph et al., 2020), and Cut-Mix (Yun et al., 2019) in Table 3. The nAP50 results show that our method consistently outperforms these augmentation methods by a large margin (+9%). This evidence demonstrates the effectiveness of our approach in the context of few-shot object detection. Detailed ablations on the number of generated images and training schemes are provided in Appendix D and Appendix E, respectively.

### 3.2.2 IS THE DIVERSITY OF A CLASS NECESSARY?

We investigate the importance of class diversity in Table 4. The table indicates that applying different augmentation techniques, which create typical and hard foregrounds, improves the performance of detectors. Specifically, controllable diffusion using ICOS in the third row diversifies prompts, enhancing the diversity of the synthetic dataset and increasing detector performance by approximately 6% nAP50 compared to not using ICOS (i.e., directly using PowerPaint with simple prompting, as shown in the first row). Additionally,

| ICOS |  | HPAS | nAP | nAP50 | nAP75 |
|------|--|------|-----|-------|-------|
| Attributes | Fine-grained. | | | | |
| | | | 34.6 | 62.1 | 34.4 |
| ✓ | | | 38.7 | 65.8 | 39.0 |
| ✓ | ✓ | | 41.2 | 68.5 | 42.6 |
| ✓ | ✓ | ✓ | 42.8 | 69.1 | 45.1 |

Table 4: Foreground-foreground ablation studies about ICOS and HPAS. nAP, nAP50, nAP75 metrics on Novel Set 1 of PASCAL VOC are reported to evaluate the importance of each modules.

by using HPAS, our method generates hard samples for FSOD, which boosts performance to 42.8%/69.1%/45.1% in nAP/nAP50/nAP75. These results demonstrate that both typical and hard foregrounds are crucial for data augmentation, especially in FSOD. Detailed ablation studies on the foreground-foreground approach are provided in Table 6, Figure 9, and Figure 10 in Appendix C.

### 3.2.3 THE IMPACT OF BACKGROUND SELECTION

In addition to studying the foreground, we also conduct ablation experiments on background selection, a crucial but often overlooked component. Table 5 demonstrates the effectiveness of background selection. With random selection, nAP, nAP50, and nAP75 metrics achieve only 34.6%, 62.1%, and 34.4%, respectively, which are lower than other background proposal strategies. By using both our typical and hard background proposal technique, the detector can be improved to 37.4%/64.1.4%/39.6% in nAP/nAP50/nAP75. These results high-

| Random | Typical | Hard | nAP | nAP50 | nAP75 |
|--------|---------|------|-----|-------|-------|
| ✓ | | | 34.6 | 62.1 | 34.4 |
| | ✓ | | 37.0 | 64.8 | 38.3 |
| | | ✓ | 35.2 | 61.9 | 36.0 |
| ✓ | | ✓ | 36.8 | 64.4 | 37.5 |
| | ✓ | ✓ | 37.4 | 64.1 | 39.6 |
| ✓ | ✓ | | 37.0 | 64.5 | 37.7 |
| ✓ | ✓ | ✓ | 36.6 | 63.6 | 38.2 |

Table 5: Foreground-background ablation study about BAP. Metrics on Novel Set 1 of PASCAL VOC are reported to evaluate the importance of each selection.

light the importance of foreground-background relations and the effectiveness of our BAP method. Therefore, we hope these types of relations can be explored more in few-shot object detection.

## 3.3 LIMITATION

There are several issues with diffusion models. The hallucinations still occur in the generated images. These circumstances can lead to parts or the entire generated object being unrelated to the prompt or resulting in low-quality synthetic images, as shown in the last two rows of Figure 8. There are several potential ways to reduce the number of hallucinations in generated data. We can apply a filter as a post-process for data generation, which can filter out objects that significantly deviate from the general characteristics. Additionally, we can apply LoRA in PEFT (Mangrulkar et al., 2022) to fine-tune the diffusion model on the few-shot data, which could generate synthetic samples with greater similarity to the current dataset and reduce hallucinations in the synthetic data. Another issue relates to the starting value $w$. This value is fixed, which may not be suitable for all novel classes.

## 4 CONCLUSION

In this paper, we introduced a novel multi-perspective data augmentation framework that enhances few-shot object detection by addressing the challenges of sample diversity and representativeness. Our approach effectively leverages in-context learning for object synthesis and incorporates a harmonic prompt aggregation scheduler to create challenging novel samples, while also improving the representation of foreground-background relationships through our Background Proposal method. Extensive experiments on several FSOD benchmarks, including PASCAL VOC, demonstrate the significant advantages of our framework over state-of-the-art methods.

## ACKNOWLEDGMENTS

This research is funded by Vietnam National University Ho Chi Minh City (VNU-HCM) under grant DS2024-26-06.

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

# A    THE RESPONSES IN ICOS

We provide ChatGPT responses in ICOS for exploring class attributes and fine-grained classes in Novel Set 1 of PASCAL VOC, as shown in Figure 6 and Figure 7, respectively.

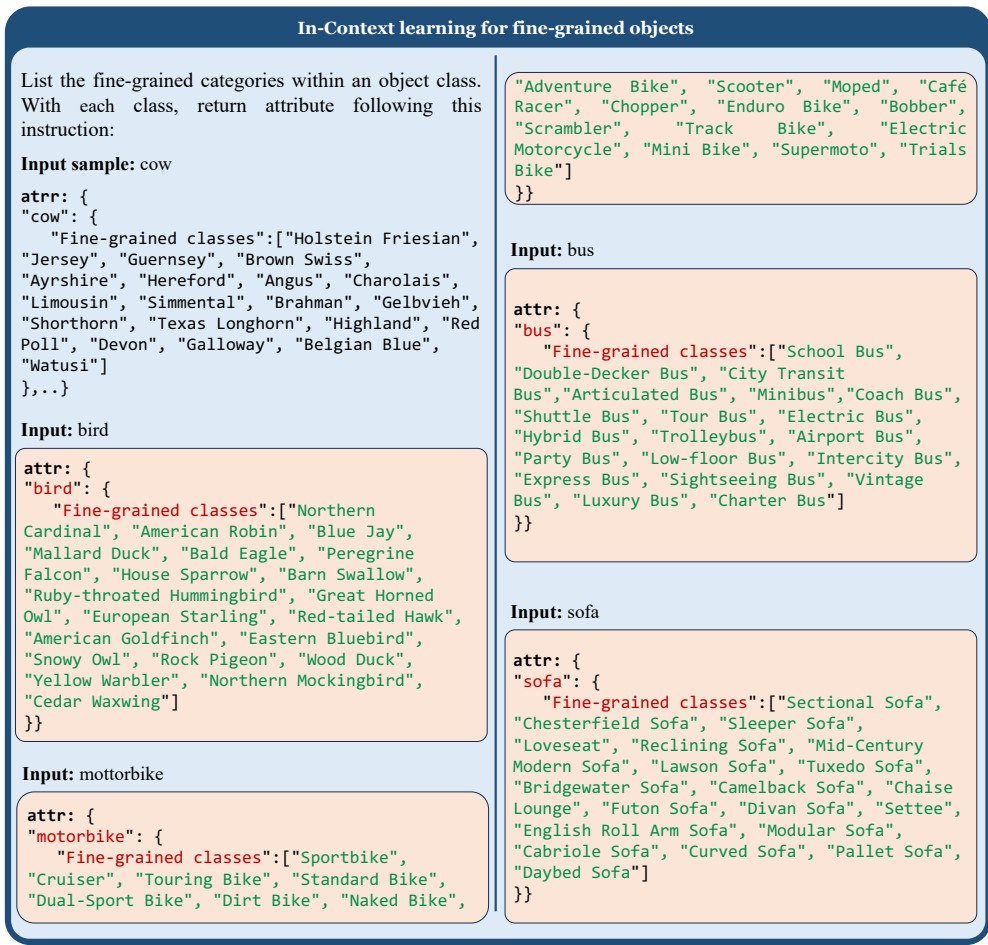

Figure 6: In-context learning results for exploring **fine-grained classes** in Novel Set 1 on the PASCAL VOC dataset.

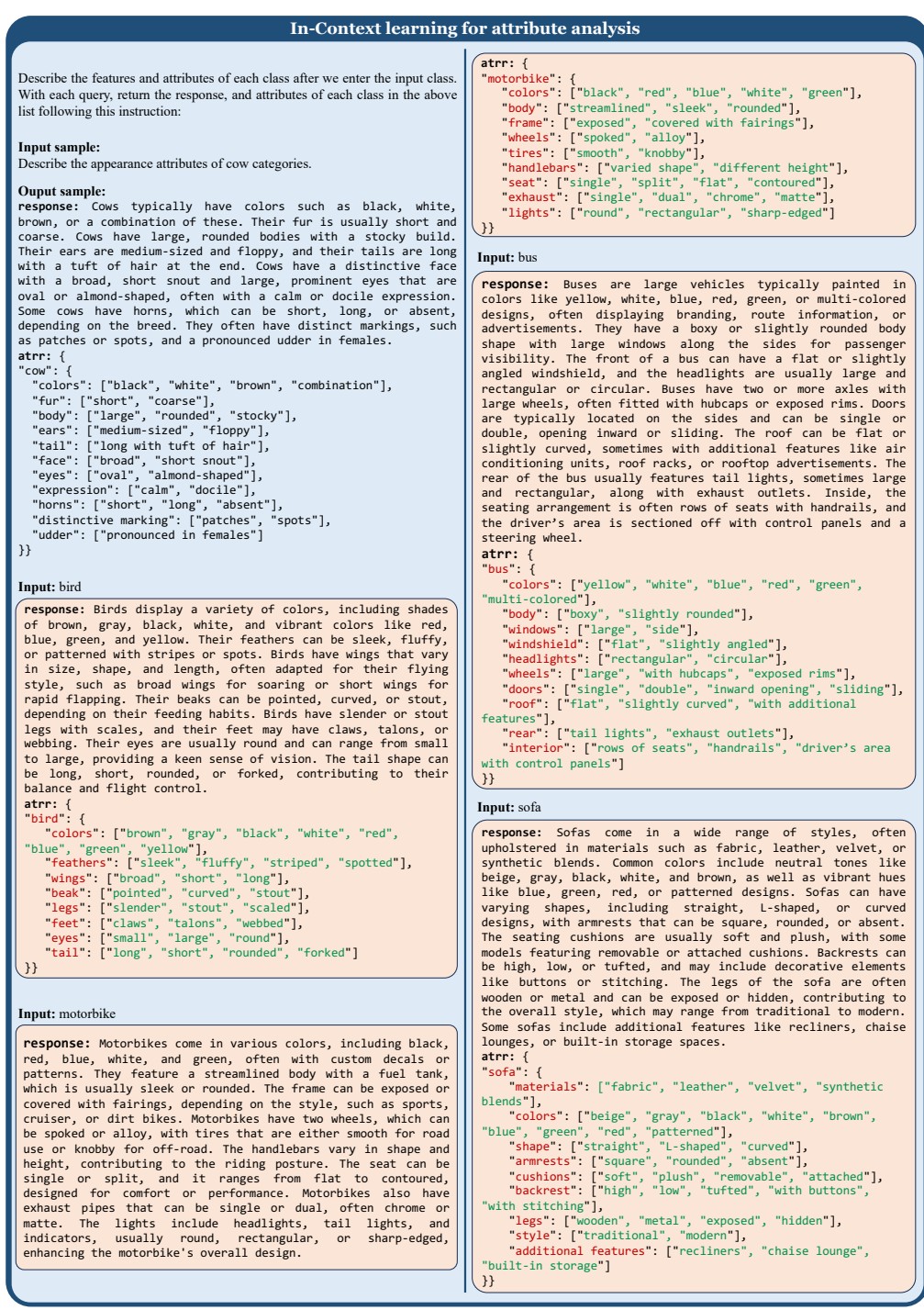

Figure 7: In-context learning results for exploring **class attributes** in Novel Set 1 on the PASCAL VOC dataset.

## B SYNTHESIS IMAGES VISUALIZATION

We provide visualization for synthesis images in Figure 8. They are created separately for each method. We visualize synthetic samples for five novel classes in Novel Set 1 on PASCAL VOC.

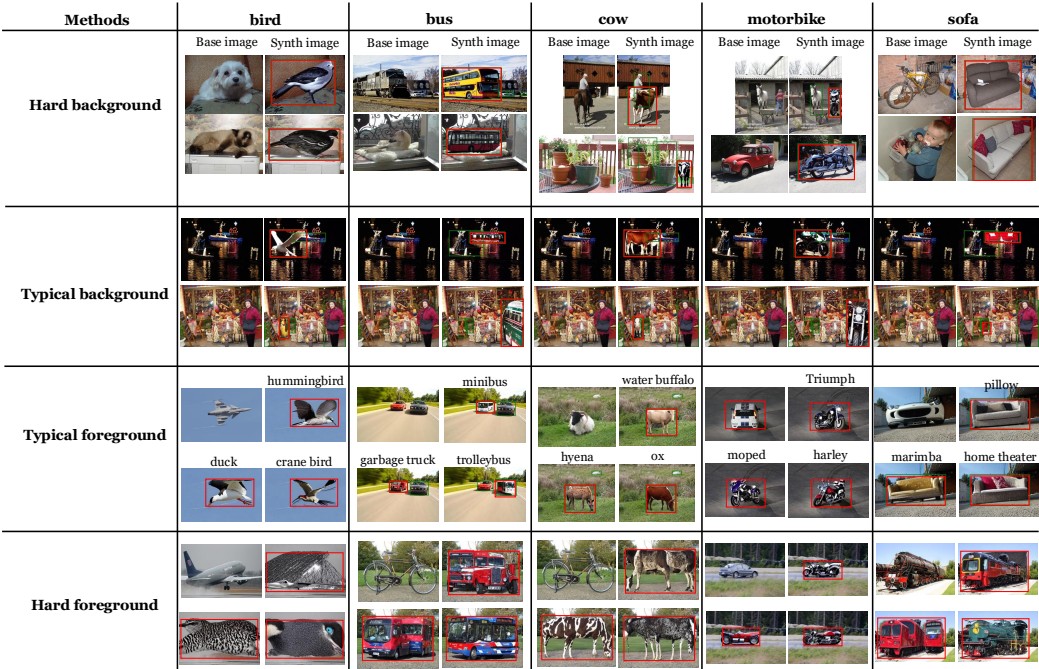

Figure 8: visualization for synthesis images. Each row represent a type of augmentation.

## C    DETAILED STUDIES FOR THE FOREGROUND-FOREGROUND APPROACH

| N.o. fine-grained classes | nAP | nAP50 | nAP75 |
|---|---|---|---|
| 1 | 40.0 | 66.8 | 42.3 |
| 2 | 39.6 | 66.3 | 40.7 |
| 3 | 39.9 | 65.4 | 43.3 |
| 4 | 41.2 | 68.5 | 42.6 |
| 5 | 40.5 | 67.1 | 42.9 |
| 6 | 38.8 | 64.8 | 39.3 |

Table 6: Ablations about the number of fine-grained classes in MPAD. nAP, nAP50, nAP75 metrics on Novel Set 1 of PASCAL VOC are reported.

As shown in Table 6 and Figure 9, we provide ablation studies in our foreground generation method. We use nAP, nAP50 and nAP75 for Table 6, and the average of these three metrics for Figure 9.

Firstly, we evaluate the impact of the number of fine-grained classes in Table 6. An upward trend is experienced when the number of fine-grained classes increases up to 4. These experiments reveal that models cannot capture datasets with very high diversity. The reason for this is that models are optimized with default training hyper-parameters (e.g., training iterations, learning rate, batch size, etc.) for FSOD. As a result, detectors are unable to converge on such datasets.

In Figure 9, we test the hyper-parameters of the HPAS method. It shows that when momentum is applied, overall performance improves. Specifically, when $m$ increases from $0.7$ to $0.9$, the model performance improves by about 1 to 2%. However, when $m$ increases to $0.99$ with a low $w$, the base class features are retained over time steps, which will generate objects too similar to the base class and reduce the model performance. Similarly, when $w$ is too low, the generated objects may retain many of the base's features. More visualization of generated samples with different $w$ and $m$ is shown in Figure 10.

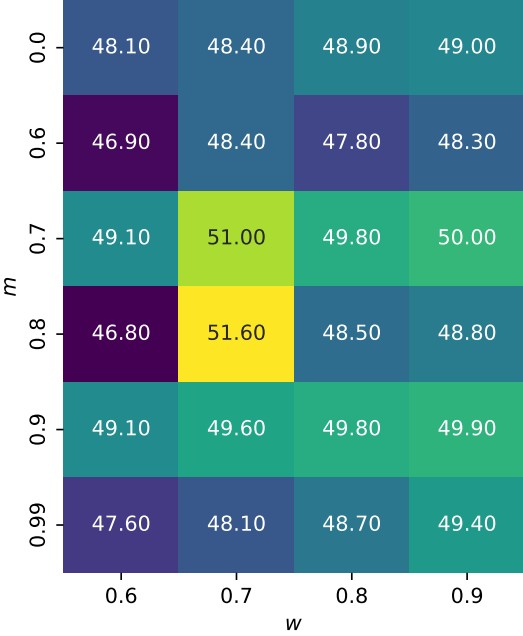

Figure 9: Ablations about momentum $m$ and starting value $w$ in Eq. (3). The average value of nAP, nAP50, nAP75 metrics on Novel Set 1 of PASCAL VOC are reported.

## D    ABLATIONS ABOUT THE NUMBER OF GENERATED IMAGES

We also provide an ablation study on the number of generated images in Table 7. These results show an upward trend in performance as the number of images increases, achieving the best performance at 300 images.

| N.o. images | 50 | 100 | 200 | **300** | 400 |
|---|---|---|---|---|---|
| nAP75 | 42.0 | 41.0 | 43.7 | **42.8** | 43.8 |
| nAP50 | 67.6 | 66.9 | 68.8 | **69.1** | 68.0 |

Table 7: Ablations about the number of generated images per class.

## E    COMPARISONS WITH DIFFERENT FINE-TUNING SCHEMES

To analyze the impact of different training schemes, we conducted experiments comparing our augmentation framework (MPAD) with alternative approaches, such as using simple prompting with ChatGPT and the diffusion model for generating training samples and fine-tuning strategies, as shown in Table 8. Specifically, we present training scheme (1), where base models are only trained on the synthetic dataset, and training scheme (2), where models are pre-trained with generated data before being fine-tuned with the original few-shot data. All models are evaluated under the 1-shot setting of Novel Set 1 of PASCAL VOC.

| #images | 5 | 10 | 50 | 100 | 200 | 300 | 400 | 600 | 800 | 1000 |
|---|---|---|---|---|---|---|---|---|---|---|
| (1) | 49.5 | 56.7 | 62.3 | 62.4 | 63.8 | 63.1 | 64.5 | 65.1 | 65.4 | 64.5 |
| (2) | 62.3 | 60.5 | 60.4 | 61.7 | 61.0 | 61.6 | 62.1 | 62.1 | 62.3 | 62.5 |

Table 8: Performance comparison between directly training on generated data (1) and fine-tuning in real data (2) using different numbers of generated samples.

The results indicate that performance saturates at 600 samples, and additional samples do not improve results. Compared to the best performance obtained in this experiment, MPAD still outperforms simple prompting by 3.7, even when using only 300 samples, as shown in Table 7.

Our training scheme is also superior to alternative pre-training and fine-tuning approaches. Specifically, MPAD effectively leverages both real high-quality few-shot samples and diverse generated samples within a single fine-tuning phase, avoiding potential overfitting issues. When pre-training with a large number of training data and fine-tuning with too few samples, the model risks "forgetting" knowledge from the pre-training phase. Moreover, MPAD is computationally efficient, requiring a simpler training process compared to alternative approaches that involve separate pre-training and fine-tuning steps.

These ablations demonstrate the effectiveness of MPAD in leveraging generated data while maintaining robustness and efficiency in training.

| Momentum | | 0.1 | 0.5 | 0.9 |
|---|---|---|---|---|
| **w** | **Class** | | | |
| **0.1** | bird | | | |
| | motobike | | | |
| | cow | | | |
| **0.7** | bird | | | |
| | motobike | | | |
| | cow | | | |
| **0.9** | bird | | | |
| | motobike | | | |
| | cow | | | |

Figure 10: Visualization of generated samples with different of momentum $(m)$ and $w$.

