# OpenReview forum: "Multi-Perspective Data Augmentation for Few-shot Object Detection"
_ICLR.cc/2025/Conference — ICLR 2025 Poster_

### Official Review · Reviewer_B8R6 · 2024-11-01

**Soundness:** 3
**Presentation:** 3
**Contribution:** 2
**Rating:** 6
**Confidence:** 4

**Summary:**

This paper proposes a Multi-Perspective Data Augmentation (MPAD) framework for few-shot object detection (FSOD), aiming to address the limitations of existing data augmentation methods. By considering both foreground-foreground and foreground-background relationships in MPAD, the authors provide an interesting solution to the problem of data augmentation. The experimental results on PASCAL VOC and MS COCO datasets demonstrate the effectiveness of the proposed methods, outperforming many state-of-the-art methods.

**Strengths:**

1. The paper is generally well-written and the idea is presented clearly.

2. The introduction provides a good background and motivation for the research

3. The proposed methods are described in detail and mostly are easy to follow.

4. The experimental setup and results are also presented in a clear and organized manner.

5. The authors also conduct a series of ablation studies to better understand the proposed method.

**Weaknesses:**

1. My main concern lies in the use of Diffusion Model (DM) and ChatGPT. From the perspective of the FSOD task, data is always the most crucial element. Consequently, it is essential to employ ChatGPT and DM to generate data as a means to address this concern, as the authors did in this study. However, the following questions arise: if ChatGPT and DM are being used, why not simply generate a substantial number of samples of the corresponding categories and conduct direct training on the generated data? Alternatively, another option could be to pre-train the detector on the generated data and subsequently perform fine-tuning on few-shot categories. Currently, the authors haven't furnished a compelling rationale to either explain, oppose, or support these viewpoints, which are of utmost significance for this paper.

    To answer these questions, I suggest the authors:

    - (1)  conduct a comprehensive analysis comparing the performance of direct training on generated data with the proposed method;

    - (2) explore the potential benefits and drawbacks of pretraining on generated data and then fine-tuning on few-shot categories. This could include experiments to determine the optimal amount of pretraining and the impact on final performance;

    - (3) provide a detailed explanation of why the proposed approach of using ChatGPT and DM in a specific way is more advantageous than (1) and (2).

2. Some details could be further elaborated. For example, in the description of the Harmonic Prompt Aggregation Scheduler (HPAS), the role of the momentum parameter and its impact on the generated samples could be explained more clearly. For example, the authors can provide a figure to visualize generated samples across a range of momentum values (e.g. 0.1, 0.5, 0.9) to illustrate how this parameter impacts the mixing of base and novel class features.

3. Additionally, the discussion on the limitations of the diffusion model in the appendix could be more in-depth, perhaps exploring potential solutions or future research directions. For example, (1) analyzing whether the hallucination will affect the final result of FSOD, and if so, what possible means can be used to solve it; or (2) is there any possible way to train (in a PEFT manner) an FSOD-aware DM to further improve the performance. In addition, I suggest putting the "limitation" part to the main text; With these changes, this paper becomes more inspiring to the reader.

**Questions:**

See "Weaknesses"

---

> ### Author Response · Authors · 2024-11-21
> **Response by the authors (1/2)**
>
> > W1: My main concern lies in the use of Diffusion Model (DM) and ChatGPT. From the perspective of the FSOD task, data is always the most crucial element. Consequently, it is essential to employ ChatGPT and DM to generate data as a means to address this concern, as the authors did in this study. However, the following questions arise: if ChatGPT and DM are being used, why not simply generate a substantial number of samples of the corresponding categories and conduct direct training on the generated data? Alternatively, another option could be to pre-train the detector on the generated data and subsequently perform fine-tuning on few-shot categories. Currently, the authors haven't furnished a compelling rationale to either explain, oppose, or support these viewpoints, which are of utmost significance for this paper. To answer these questions, I suggest the authors: (1) conduct a comprehensive analysis comparing the performance of direct training on generated data with the proposed method; (2) explore the potential benefits and drawbacks of pretraining on generated data and then fine-tuning on few-shot categories. This could include experiments to determine the optimal amount of pretraining and the impact on final performance; (3) provide a detailed explanation of why the proposed approach of using ChatGPT and DM in a specific way is more advantageous than (1) and (2).
>
> We thank the reviewer for your critical comment. We completely agree that data is the most crucial issue in the FSOD task. To ensure we understand your comment correctly, we restate the following points:
>
> **Why our augmentation framework (MPAD) is more advantageous than using simple prompting with ChatGPT and DM. Could using simple prompting with a substantial number of samples to train the model be better than our method?**
>
>
>
> The main difference between our MPAD and previous works (including simple prompting) is that we leverage the foreground-background relation. Simple prompting (e.g., using ChatGPT to generate prompts for DM to create images) results in a lack of diverse and hard samples for training the model. In object detection task, the background plays a crucial role. Using simple prompting, ChatGPT and DM generate data with uncontrolled backgrounds.
>
> | #images |    5 |   10 |   50 |  100 |  200 |  300 |  400 |  600 |  800 | 1000 |
> |---------|-----:|-----:|-----:|-----:|-----:|-----:|-----:|-----:|-----:|-----:|
> | (1)     | 49.5 | 56.7 | 62.3 | 62.4 | 63.8 | 63.1 | 64.5 | 65.1 | 65.4 | 64.5 |
> | (2)     | 62.3 | 60.5 | 60.4 | 61.7 | 61.0 | 61.6 | 62.1 | 62.1 | 62.3 | 62.5 |
>
> To confirm this point, we conducted additional experiments, and the results are reported in the table above. We simply generated data with varying sample sizes: 5, 10, 50, 100, 200, 300, 400, 600, 800, and 1000, then trained the model. The performance becomes saturated when the number of samples reaches 600 (i.e. performance does not improve when increasing #generated samples). Compared to the best performance in this experiment, our model still outperforms by $3.7\%$, even using only 300 samples. We also added the experiments to determine the number of generated images in Table 7 in the Appendix section.
>
> **Why our training scheme using both generated data and original few-shot data is better than (1) training with generated data only, and (2) pretraining with generated data, then fine-tuning with original few-shot data?**
>
> Our training scheme is superior to (1) because it leverages both few-shot samples (real and high-quality samples) and diverse generated samples in a single training time. Our MPAD is less prone to overfitting compared to scheme (2). Even when pretrained with a large amount of training data, fine-tuning with too few samples can cause the model to "**forget**" knowledge from the pretraining phase. Moreover, MPAD is less computationally expensive due to its simpler training process, while (2) requires training twice.
>
> To confirm this point, we conducted further experiments with the same number of images. Using scheme (2), the performance significantly decreased to $61.6\%$, compared to $63.1\%$ of scheme (1) and $69.1\%$ of our MPAD.

---

> > ### Author Response · Authors · 2024-11-21
> > **Response by the authors (2/2)**
> >
> > > W2: Some details could be further elaborated. For example, in the description of the Harmonic Prompt Aggregation Scheduler (HPAS), the role of the momentum parameter and its impact on the generated samples could be explained more clearly. For example, the authors can provide a figure to visualize generated samples across a range of momentum values (e.g. 0.1, 0.5, 0.9) to illustrate how this parameter impacts the mixing of base and novel class features.
> >
> > We thank the reviewer for the constructive review. HPAS is designed to generate hard samples by blending the features of two classes. For quantitative results, as shown in Figure 9 in appendix section, when the momentum is applied, the overall performance of FSOD models showed improvement. However, while $m=0.99$, it still remains too much base features and creates harder samples, which leads to the performance decrease. For qualitative results, we updated and provided additional visualizations of samples with different $m$ and $w$ in Figure 10 in the Appendix section.
> >
> > > W3: Additionally, the discussion on the limitations of the diffusion model in the appendix could be more in-depth, perhaps exploring potential solutions or future research directions. For example, (1) analyzing whether the hallucination will affect the final result of FSOD, and if so, what possible means can be used to solve it; or (2) is there any possible way to train (in a PEFT manner) an FSOD-aware DM to further improve the performance. In addition, I suggest putting the "limitation" part to the main text; With these changes, this paper becomes more inspiring to the reader.
> >
> > We thank the reviewer for the constructive suggestion. We updated and moved the limitation to the main text (lines 468-475, page 9): "There are several issues with diffusion models. The hallucinations still occur in the generated images. These circumstances can lead to parts or the entire generated object being unrelated to the prompt or resulting in low-quality synthetic images, as shown in the last two rows of Figure 8. There are several potential ways to reduce the number of hallucinations in generated data. We can apply a filter as a post-process for data generation, which can filter out objects that significantly deviate from the general characteristics. Additionally, we can apply LoRA in PEFT Mangrulkar et al. (2022) to fine-tune the diffusion model on the few-shot data, which could generate synthetic samples with greater similarity to the current dataset and reduce hallucinations in the synthetic data. Another issue relates to the starting value $w$. This value is fixed, which may not be suitable for all novel classes."
> >
> > **We hope that our response has fully addressed your concerns. Please kindly let us know if you need any further clarification or information.**

---

> > > ### Comment · Reviewer_B8R6 · 2024-11-24
> > > **Post-Rebuttal**
> > >
> > > Thanks for the response from the authors. I have carefully checked the response from the authors, and most of my concerns have been addressed. I am happy to maintain my initial score.
> > >
> > > PS: I suggest to put more analysis about the above two schemes for FSOD in the main text. This would help readers better understand the advantage of the proposed method.

---

> ### Author Response · Authors · 2024-11-25
> **Response to Reviewer B8R6**
>
> We deeply appreciate reviewer B8R6 for valuable time and providing insightful feedback that has enhanced our manuscript. We will include the analysis of various training schemes in our manuscript.

---

### Official Review · Reviewer_jFnB · 2024-11-02

**Soundness:** 3
**Presentation:** 3
**Contribution:** 3
**Rating:** 6
**Confidence:** 5

**Summary:**

This paper proposes a Multi-Perspective Data Augmentation (MPAD) framework aimed at enhancing few-shot object detection (FSOD) by generating diverse and challenging synthetic samples. The MPAD framework utilizes techniques such as Chain-of-Thought Prompting for Object Synthesis (CPOS), Harmonic Prompt Aggregation Scheduler (HPAS), and Background Proposal (BAP) to create representative and hard samples. The authors present results demonstrating significant performance improvements on PASCAL VOC and MS COCO benchmarks, showing an average increase of 17.5% in nAP50 over baseline methods.

**Strengths:**

1.	The CPOS and HPAS introduce novel ways to leverage both typical and hard samples, leading to a more representative synthetic dataset.
2.	The use of BAP to generate diverse backgrounds helps enhance detection accuracy by allowing the model to distinguish between foreground and background more effectively.
3.	The proposed framework achieves notable gains over state-of-the-art baselines on multiple FSOD benchmarks, particularly in challenging low-shot settings.

**Weaknesses:**

1. The framework combines multiple advanced techniques, including diffusion models, harmonic prompt scheduling, and complex background sampling, which may make it challenging for practitioners to implement effectively in real-world scenarios. At the same time, it increases the complexity of the model. Please analyze the model complexity and real-time inference of the generated model.
2. The paper demonstrates performance gains, but it would benefit from a more granular analysis comparing the effectiveness of each augmentation component (CPOS, HPAS) against similar elements in other FSOD methods. Has the pre-trained model of the generated model already seen various categories to generate more realistic new class samples when regenerated again? Can other generative models also generate samples to improve the performance of small samples, and does the proposed method have such a significant performance improvement compared to this type of method
Data Efficiency in Synthetic Generation:
3. While the method performs well, additional experiments assessing data efficiency could be valuable. Evaluating how much synthetic data is optimal or exploring the impact of different amounts of augmented data on performance could provide insights into the scalability and efficiency of the framework.

**Questions:**

Please refer to the Weaknesses box.

---

> ### Author Response · Authors · 2024-11-21
> **Response by the authors**
>
> > W1: The framework combines multiple advanced techniques, including diffusion models, harmonic prompt scheduling, and complex background sampling, which may make it challenging for practitioners to implement effectively in real-world scenarios. At the same time, it increases the complexity of the model. Please analyze the model complexity and real-time inference of the generated model.
>
> Our method adjusts feature embeddings in a controllable diffusion process without making any modifications to the model architecture. Specifically, instead of using text embeddings as described in PowerPaint, we use $\gamma_c^t$ as defined in Equation (3) (line 258, page 5). Additionally, base features are precomputed at the start of the generation process and computed only once. Therefore, there is no significant increase in computational cost or complexity. The inference time for the generation process primarily depends on the inference time of the diffusion model and the number of generated images.
>
> > W2: The paper demonstrates performance gains, but it would benefit from a more granular analysis comparing the effectiveness of each augmentation component (CPOS, HPAS) against similar elements in other FSOD methods. Has the pre-trained model of the generated model already seen various categories to generate more realistic new class samples when regenerated again? Can other generative models also generate samples to improve the performance of small samples, and does the proposed method have such a significant performance improvement compared to this type of method Data Efficiency in Synthetic Generation:
>
> We use the pretrained PowerPaint on Open Images [3] and LAION-Aesthetics V2 5+ [4]. Therefore, all novel and base classes are seen during pretraining. We also compare our method with FSOD approaches using other generative models, as shown in Table 1 and Table 2. Specifically, [1] uses a VAE to generate synthetic features, while [2] uses an MAE to reconstruct new images for novel classes. Our method surpasses [1] and [2] by approximately 15% and 11%, respectively.
>
> > W3: While the method performs well, additional experiments assessing data efficiency could be valuable. Evaluating how much synthetic data is optimal or exploring the impact of different amounts of augmented data on performance could provide insights into the scalability and efficiency of the framework.
>
>
> | #images |   50 |  100 |  200 |  300 |  400 |
> |---------|-----:|-----:|-----:|-----:|-----:|
> | nAP75   | 42.0 | 41.0 | 43.7 | **42.8** | 43.8 |
> | nAP50   | 67.6 | 66.9 | 68.8 | **69.1** | 68.0 |
>
>
> We thank the reviewer for the constructive review. We added the experiments to determine the number of generated images in Table 7 in the Appendix. We chose the number of generated images to be 300 for best results.
>
> **References**:
>
> [1] Few-Shot Object Detection via Variational Feature Aggregation, AAAI 2023.
>
> [2] SNIDA: Unlocking Few-Shot Object Detection with Non-linear Semantic Decoupling Augmentation, CVPR 2024.
>
> [3] Unified image classification, object detection, and visual relationship detection at scale, IJCV 2020.
>
> [4] Laion-5b: An open large scale dataset for training next generation image-text models, NeurIPS 2022.
>
> **We hope that our response has fully addressed your concerns. Please kindly let us know if you need any further clarification or information.**

---

> > ### Comment · Reviewer_jFnB · 2024-11-27
> >
> > After checking the responses to my questions, I believe that most of my concerns have been resolved. Considering the originality and quality of this article, I decided to maintain my rating as “ 6: marginally above the acceptance threshold.”

---

> > > ### Author Response · Authors · 2024-11-28
> > > **Official Comment by Authors**
> > >
> > > We sincerely thank Reviewer jFnB very much for reading our responses. We deeply appreciate the reviewer for valuable time and for providing insightful feedback.

---

### Official Review · Reviewer_iPh4 · 2024-11-03

**Soundness:** 2
**Presentation:** 2
**Contribution:** 2
**Rating:** 6
**Confidence:** 5

**Summary:**

This paper attempts to enhance the diversity of data generation by analyzing typical and hard samples. Specifically, a data generation architecture is proposed that comprehensively considers the typicality and difficulty of foreground and background selections, establishing the relationship between foreground and background. Experimental evidence demonstrates significant performance improvements compared to recent methods.

**Strengths:**

+ From the experimental data, the current data augmentation methods show a certain degree of performance improvement.
+ Integrating various mainstream generative models and zero-shot learning models, including diffusion, CLIP, etc.

**Weaknesses:**

- The present approach heavily depends on utilizing pre-trained models to select typical and challenging samples, potentially causing interference when assessing the efficacy of augmentation strategies.
- Additional clarification is needed regarding the fairness of the experiments.

**Questions:**

1. What are the definitions of typical and hard samples? Detailed assessment criteria need to be further elaborated.
2. In Section 2.2, lines 159-161 mention that HPAS can generate hard samples. Strategically, it may indeed involve a wider range of categories. However, for the model, the level of difficulty in recognition is not solely dependent on the number of categories in the image. Why are samples generated based on mixed prompt embedding considered hard samples in HPAS?
3. In the last paragraph on the fifth page, it mentions the concept of camouflage targets. Camouflaged objects typically refer to objects where the foreground and background have high similarity. However, in the subsequent text, the selection of data with high similarity between the background in the base stage and the classes in the novel stage as challenging data does not explicitly demonstrate the connection between selecting difficult backgrounds and camouflaged objects. Therefore, what is the relationship between the selection of challenging backgrounds and camouflaged objects?
4. In Equation 6, it is mentioned that typical clutter background is obtained by calculating entropy. Such selection heavily relies on the performance of the feature extractor F. If the predictive results are good, few background images may be selected. However, if the predictive results are poor, do the selected background images hold any useful value?
5. The architecture of this paper incorporates various methods with pre-trained models, such as CLIP. Is the fairness of the comparison between the current augmentation strategy and previous methods lacking?

---

> ### Author Response · Authors · 2024-11-21
> **Response by the authors (1/2)**
>
> > W1: The present approach heavily depends on utilizing pre-trained models to select typical and challenging samples, potentially causing interference when assessing the efficacy of augmentation strategies
>
> We follow previous works [1, 2, 3, 4, 5, 6, 7] to use the pretrained models in our method. We also compare FSOD methods using pretrained model, which is clarified in the below response.
>
> > W2: Additional clarification is needed regarding the fairness of the experiments.
>
> We thank the reviewer for positive suggestion. We clarified the fairness of the experiments and updated in the manuscript in the Experiments section (lines 413-417, page 8): "Considerably, our MPAD surpasses previous works Wang et al. (2024); Lin et al. (2023); Kaul et al. (2022); Li et al. (2023a); Zhu et al. (2021); Xu et al. (2023) that use pretrained CLIP, ViT, diffusion models, language models, or post-processing in detection. Meanwhile, methods Wang et al. (2024); Lin et al. (2023) are state-of-the-art data augmentation methods in FSOD"
>
> > Q1: What are the definitions of typical and hard samples? Detailed assessment criteria need to be further elaborated
>
> In this paper, we assume that typical samples contain only features of one class and are created using augmented prompts with a diffusion model. Hard foregrounds are blended with features of two classes using HPAS. We also emphasize this point in the introduction section (lines 071-075, page 1): "In other words, in this paper, we define typical samples as those that contain features of a single class, whereas hard samples exhibit features of two classes."
>
>
> > Q2: In Section 2.2, lines 159-161 mention that HPAS can generate hard samples. Strategically, it may indeed involve a wider range of categories. However, for the model, the level of difficulty in recognition is not solely dependent on the number of categories in the image. Why are samples generated based on mixed prompt embedding considered hard samples in HPAS?
>
> We agree with the review. There are other transformations that can create hard samples, such as occlusion and scale. These types of transformations are embedded during the random selection of bounding boxes with multiple scales and positions. In this paper, we consider another type of hard sample: ambiguous objects that contain features of more than one class and are hard to recognize. To generate these types of hard samples, we propose using HPAS. To the best of our knowledge, this is the first work that considers this type of hard sample in data augmentation for FSOD.
>
> > Q3: In the last paragraph on the fifth page, it mentions the concept of camouflage targets. Camouflaged objects typically refer to objects where the foreground and background have high similarity. However, in the subsequent text, the selection of data with high similarity between the background in the base stage and the classes in the novel stage as challenging data does not explicitly demonstrate the connection between selecting difficult backgrounds and camouflaged objects. Therefore, what is the relationship between the selection of challenging backgrounds and camouflaged objects?
>
> To define the hard background, we are inspired by the concept of camouflaged objects, where the foreground and background have high similarity in appearance. To simulate that phenomenon, we encode the visual information of the background (from the base set) and a class representative (from the synthetic dataset), then compare them using a similarity metric like cosine similarity. We do not create a camouflage dataset; we simply inherit the concept of visual similarity between foreground and background.
>
> > Q4: In Equation 6, it is mentioned that typical clutter background is obtained by calculating entropy. Such selection heavily relies on the performance of the feature extractor F. If the predictive results are good, few background images may be selected. However, if the predictive results are poor, do the selected background images hold any useful value?
>
> We agree with the reviewer. A poor $F$ extractor may evaluate all base images at the same point and degrade the performance of MPAD. Therefore, we choose a general and good enough extractor (ViT) which is also used in several previous works [2,3,4,5] in FSOD.

---

> > ### Author Response · Authors · 2024-11-21
> > **Response by the authors (2/2)**
> >
> > > Q5: The architecture of this paper incorporates various methods with pre-trained models, such as CLIP. Is the fairness of the comparison between the current augmentation strategy and previous methods lacking?
> >
> > We thank the reviewer for positive suggestion. We have compared our method with other approaches using either pretrained CLIP or diffusion models in Table 1, Table 2, and Table 4 in the manuscript. In Table 1, our method achieves better performance than [1, 6, 7]. While [6, 7] use pretrained CLIP in their methods, [1] leverages both pretrained CLIP and a pretrained diffusion model to generate synthetic samples for novel classes. Additionally, in Table 4, when compared to the baseline (using only the diffusion model, PowerPaint) in the first row, our method in the last row outperforms it by 7% overall.
> >
> >
> > **References**:
> >
> > [1] Explore the Power of Synthetic Data on Few-shot Object Detection, CVPRW 2023.
> >
> > [2] Label, Verify, Correct: A Simple Few Shot Object Detection Method, CVPR 2022.
> >
> > [3] Proposal Distribution Calibration for Few-Shot Object Detection, TCSVT 2023.
> >
> > [4] Mask-Guided Vision Transformer for Few-Shot Learning, TCSVT 2024.
> >
> > [5] Few-Shot Object Detection with Foundation Models, CVPR 2024.
> >
> > [6] Disentangle and remerge: interventional knowledge distillation for few-shot object detection from a conditional causal perspective, AAAI 2023.
> >
> > [7] SNIDA: Unlocking Few-Shot Object Detection with Non-linear Semantic Decoupling Augmentation, CVPR 2024.
> >
> > **We hope that our response has fully addressed your concerns. Please kindly let us know if you need any further clarification or information.**

---

### Official Review · Reviewer_RKTD · 2024-11-03

**Soundness:** 3
**Presentation:** 2
**Contribution:** 1
**Rating:** 6
**Confidence:** 5

**Summary:**

This paper presents a Multi-Perspective Data Augmentation (MPAD) framework to improve few-shot object detection (FSOD) by generating diverse and representative synthetic samples. The framework includes three components: Chain-of-Thought Prompting for Object Synthesis (CPOS) uses large language models to enhance prompt diversity with fine-grained attributes; Harmonic Prompt Aggregation Scheduler (HPAS) mixes base and novel class features in the diffusion process, producing hard-to-classify samples; and Background Proposal (BAP) selects complex or visually similar backgrounds to improve object-background differentiation. Extensive experiments on PASCAL VOC and COCO few-shot object detection benchmarks show the effectiveness of the proposed method.

**Strengths:**

1: The MPAD framework leverages Chain-of-Thought Prompting to generate prompts with fine-grained attributes, enabling diverse and representative data synthesis for few-shot object detection.

2: The proposed method is easy to follow and not limited to some specific object detection architectures, making the technique be applied in different scenarios without further modifications.

3: The main results on PASCAL VOC and COCO few-shot datasets illustrate that the proposed method can improve non-trivial performance on many few-shot object detection benchmarks.

**Weaknesses:**

1: The description that "this work is the first ..." in Lines 81-83 is a little bit over-claimed. From my perspective, there is already a large amount of work exploring using large-scale pretrained diffusion models for object detection by generation (like general object detection and corner case generation for autonomous driving). Simply extending the setting to few-shot object detection isn't that significant to me.

2: Section 2.3 (CoT prompting for object synthesis) actually belongs to prompt engineering. The Chain-of-Thought emphasizes solving a problem step by step. For object synthesis discussed in this paper, the task simply needs to list all possible attributes of a category ( or use a pre-defined template). This can be regarded as in-context learning or prompt engineering rather than Chain-of-Thought. Meanwhile, the inpainting model is also modified from other previous works.

3: The overall technique contribution is limited. Continuing from point 2, the method is more like a combination of several existing verified techniques for object detection data generation without a clear logical chain. The HPAS is a prompt embedding level mixup data augmentation which is also not novel (the insight of embedding mixup is common in conditional data generation using diffusion model). It's also hard to convince me that the BAP part brings much novelty.

4: Comparisons with deep learning methods (using large-scale pretrained models like diffusion model and CLIP, etc). In Table 3, the authors only compare their proposed method with some non-deep learning-based methods. However, given the widely used conditional diffusion model for object detection object generation, the authors also should add more baselines such as simply using the **Powerpoint** inpainting model for data generation, to further verify the effectiveness of the proposed module that is claimed to enhance the diversity.

Overall, I tend to vote for rejection at this time because of the technical contribution and the proposed method itself, along with comprehensive comparisons. I am waiting and will hear from other reviewers.

**Questions:**

1: In the Introduction part (especially for Figure 1), what's the definition of typical and hard objects on the novel set? How to distinguish them?

2: Which are the novel synthetic samples and the base real samples in Novel Set 1 in Figure 1? Why do some colors (classes) only have solid circle samples without any "x" samples? (I am deeply Confused by Figure 1.)

3: In Table 2, is it a typo that sometimes the metric is "nAP" while sometimes not?

**Details Of Ethics Concerns:**

N/A.

---

> ### Author Response · Authors · 2024-11-21
> **Response by the authors (1/3)**
>
> We sincerely thank the reviewer for the constructive comments. There are several previous works [1, 2] using diffusion models in FSOD. However, such works do not focus on exploiting the foreground-background relation or extending the ability of diffusion models by using LLMs. To our knowledge, this is the first work to use ChatGPT to diversify prompts and embed the foreground-background relations when synthesizing diverse datasets in few-shot object detection.
>
> > W1: The description that "this work is the first ..." in Lines 81-83 is a little bit over-claimed. From my perspective, there is already a large amount of work exploring using large-scale pretrained diffusion models for object detection by generation (like general object detection and corner case generation for autonomous driving). Simply extending the setting to few-shot object detection isn't that significant to me.
>
> As we mentioned in the above response, this is the first work using ChatGPT to diversify prompts and exploiting the foreground-background relations. Prior FSOD works [1,2] only use diffusion models with the simple prompts. We also show that simply using diffusion model is not enough to create diverse datasets in Table 4. Specifically, our MPAD (last row) can improve over 7% for all metrics in comparison with straightforward method using only controllable diffusion method (first row).
>
> We also emphasize this point in the Experiments section (lines 442-443, page 9) : "Specifically, controllable diffusion using ICOS in the third row diversifies prompts, enhancing the diversity of the synthetic dataset and increasing detector performance by approximately $6\%$ nAP50 compared to not using ICOS (i.e., directly using PowerPaint with simple prompting, as shown in the first row)"
>
>
> > W2: Section 2.3 (CoT prompting for object synthesis) actually belongs to prompt engineering. The Chain-of-Thought emphasizes solving a problem step by step. For object synthesis discussed in this paper, the task simply needs to list all possible attributes of a category ( or use a pre-defined template). This can be regarded as in-context learning or prompt engineering rather than Chain-of-Thought. Meanwhile, the inpainting model is also modified from other previous works.
>
> We thank the reviewer for your valuable suggestion. Both CoT and in-context learning share attributes of prompt engineering, such as providing detailed information in the input. We have revised "CoT" to "in-context learning" and updated the manuscript accordingly.

---

> > ### Author Response · Authors · 2024-11-21
> > **Response by the authors (2/3)**
> >
> > > W3: The overall technique contribution is limited. Continuing from point 2, the method is more like a combination of several existing verified techniques for object detection data generation without a clear logical chain. The HPAS is a prompt embedding level mixup data augmentation which is also not novel (the insight of embedding mixup is common in conditional data generation using diffusion model). It's also hard to convince me that the BAP part brings much novelty.
> >
> >
> > The main point of this comment is about our novelty. Here are some points to highlight:
> >
> > * **This work is new**. Previous works [3, 4] only use LLMs to create class embeddings and do not exploit prior knowledge to enrich the set of attributes of novel classes. Some other works [1,2] solely utilize diffusion models with simple prompts to create datasets that lack diversity. Our framework uses ChatGPT with in-context learning to discover class attributes (e.g., class parts, color, shape). Additionally, previous works are not aware of foreground-background relations to create highly representative samples (i.e., typical and hard samples). [1, 2] only use random background which leads to the mismatch between the foreground and the context. Our MPAD is the first work to address these concerns, as mentioned in the above response.
> >
> > * **This work is non-trivial**: Our goal is to generate highly representative and diverse training data. Based on the large margin principle, boundary samples play a crucial role in forming classifiers. We consider these samples as hard samples (objects that have attributes of more than one class). In contrast, typical samples formulate the general attributes for each class.
> >
> > To create typical and hard backgrounds, we propose BAP to collect highly similar and clustered backgrounds from the base dataset. Experimental results indicate that our BAP boosts performance by ~$3\%$ compared to using random backgrounds.
> >
> > To create typical foregrounds, we use attributes from ICOS to diversify typical samples. To create hard foregrounds, we propose HPAS to mix features of two classes in the embedding space with adaptive weights **instead of using the fixed weight like previous works [6, 7]**. Specifically, we leverage the property of the reverse diffusion process to propose a weighted scheduler to blend two class features. At denoising timestep $t = T$, the base weights account for a reasonable proportion to generate the object with low-level features of the base class, and then gradually decrease to $0$ at $t = 0$ to generate high-level features of the novel class.
> >
> > Last but not least, our framework does not need to fine-tune any LLM models, which may lead to overfitting problems. We only use a pretrained diffusion model and an LLM model (e.g., ChatGPT) for generic data generation.
> >
> >
> > > W4: Comparisons with deep learning methods (using large-scale pretrained models like diffusion model and CLIP, etc). In Table 3, the authors only compare their proposed method with some non-deep learning-based methods. However, given the widely used conditional diffusion model for object detection object generation, the authors also should add more baselines such as simply using the Powerpoint inpainting model for data generation, to further verify the effectiveness of the proposed module that is claimed to enhance the diversity.
> >
> > We thank the reviewer for the constructive comments. We compared our method with other deep learning approaches (using large-scale pretrained models) on FSOD, as shown in Table 1 and Table 2. Specifically, MPAD significantly outperforms other methods using pretrained CLIP [3, 4, 5] and pretrained diffusion models [1]. We have revised to add this point in the Experiments section (lines 413-417, page 8): "Considerably, our MPAD surpasses previous works Wang et al. (2024); Lin et al. (2023); Kaul et al. (2022); Li et al. (2023a); Zhu et al. (2021); Xu et al. (2023) that use pretrained CLIP, ViT, diffusion models, language models, or post-processing in detection. Meanwhile, methods Wang et al. (2024); Lin et al. (2023) are state-of-the-art data augmentation methods in FSOD"
> >
> > Additionally, we provided a comparison between the baseline model (first row), which simply uses PowerPaint, and our proposed method (last row) in Table 4 (lines 441–443, page 9). These results demonstrate our significant improvement in FSOD.

---

> > > ### Author Response · Authors · 2024-11-21
> > > **Response by the authors (3/3)**
> > >
> > > > Q1: In the Introduction part (especially for Figure 1), what's the definition of typical and hard objects on the novel set? How to distinguish them?
> > >
> > > Our assumption is that typical samples only contain features of one class, while hard samples have features of more than one class. Typical samples are generated by ICOS, while hard ones are generated using HPAS. We also emphasize this point in the introduction section (lines 071-075, page 1): "In other words, in this paper, we define typical samples as those that contain features of a single class, whereas hard samples exhibit features of two classes."
> > >
> > > > Q2: Which are the novel synthetic samples and the base real samples in Novel Set 1 in Figure 1? Why do some colors (classes) only have solid circle samples without any "x" samples? (I am deeply Confused by Figure 1.)
> > >
> > > We thank the reviewer for pointing this out. We have revised it to include a more detailed caption in the new version. In Figure 1 (lines 072–080, page 2): "T-SNE visualization of novel synthetic typical/hard samples and base real samples in Novel Set 1 of PASCAL VOC. We only generate synthetic samples for three novel classes (bird, bus, cow) and use real samples for three base classes (aeroplane, train, horse). Typical and hard samples in novel classes are created by using ICOS and HPAS, respectively. Base real samples are considered as typical samples."
> > >
> > > > Q3: In Table 2, is it a typo that sometimes the metric is "nAP" while sometimes not?
> > >
> > > We thank the reviewer for the positive comment. We proofread and updated the manuscript.
> > >
> > > **References**:
> > >
> > > [1] Explore the Power of Synthetic Data on Few-shot Object Detection, CVPRW 2023.
> > >
> > > [2] Data Augmentation for Object Detection via Controllable Diffusion Models, WACV 2024.
> > >
> > > [3] Semantic Relation Reasoning for Shot-Stable Few-Shot Object Detection, CVPR 2021.
> > >
> > > [4] Disentangle and remerge: interventional knowledge distillation for few-shot object detection from a conditional causal perspective, AAAI 2023.
> > >
> > > [5] SNIDA: Unlocking Few-Shot Object Detection with Non-linear Semantic Decoupling Augmentation, CVPR 2024.
> > >
> > > [6] Imagic: Text-Based Real Image Editing with Diffusion Models, CVPR 2023.
> > >
> > > [7] Forgedit: Text-guided Image Editing via Learning and Forgetting, arXiv 2024.
> > >
> > > **We hope that our response has fully addressed your concerns. Please kindly let us know if you need any further clarification or information.**

---

> ### Author Response · Authors · 2024-11-28
> **Official Comment by Authors**
>
> Dear Reviewer RKTD,
>
> We thank the Reviewer for your time and effort in evaluating our paper and providing valuable feedback. We have carefully addressed your comments and incorporated improvements in our rebuttal. If you have had a chance to review our response, we would greatly appreciate your thoughts. Additionally, we sincerely hope you will consider updating your score if you feel our revisions have effectively addressed your concerns.

---

> ### Author Response · Authors · 2024-12-02
> **Official Comment by Authors**
>
> Dear Reviewer RKTD,
>
> As the extended discussion period nears its conclusion, we would greatly appreciate it if you could take a moment to review our rebuttals and let us know if you have any further questions or concerns. Our goal is to address all concerns thoroughly during the discussion phase, and your feedback is vital to this process.

---

### Author Response · Authors · 2024-11-21
**General response**

We are very grateful to the reviewers for carefully reviewing our paper and providing constructive comments and suggestions.

We would like to emphasize that this paper addresses the problems with current augmentation methods, which fail to simulate both typical and challenging samples, crucial for capturing the data's diversity. Current works imply generate samples using very basic prompts and transformations, and especially lack awareness of the relation between background and foreground. Our proposed framework, MPAD, leverages the power of LLMs (e.g., ChatGPT) to embed prior knowledge into prompts and exploit background-foreground relations for synthesizing novel images. We have revised the paper carefully according to the comments and suggestions, where the changed parts are colored in blue. Our response to individual reviewers can be found in the personal replies, but we would also like to provide a brief summary of revisions for your convenience.

- We have revised Chain-of-Thought prompting to in-context learning and **CPOS** to **ICOS**. We have updated the manuscript accordingly.

- We have clarified the fairness of the experiments and highlighted the previous few-shot object detection (FSOD) works using pretrained CLIP, language models, or diffusion models.


**We hope that our response has fully addressed your concerns. Please kindly let us know if you need any further clarification or information.**

---

### Meta-Review · Area_Chair_Yscr · 2024-12-22

**Metareview:**

This paper proposes the Multi-Perspective Data Augmentation (MPAD) framework to enhance few-shot object detection (FSOD) by generating diverse and representative synthetic samples. The initial scores for the paper were mixed. After the rebuttal, two reviewers maintained their acceptance scores, while two others revised their scores from rejection to acceptance. Ultimately, all four reviewers reached a consensus to accept the paper. After carefully reviewing the comments and rebuttals, the Area Chair (AC) concurs with the merits presented in the paper and recommends its acceptance. However, the authors are advised to update the final version in line with the suggestions provided by all reviewers.

**Additional Comments On Reviewer Discussion:**

Four reviewers reviewed this paper with thorough comments, e.g., on representation issues, technique contributions, writing/figure issues, etc. After rebuttal, most reviewers assumed their concerns had been addressed. AC also agrees with the merits of this paper.

---

### Decision · Program_Chairs · 2025-01-22

Accept (Poster)